

# Post-Caledonian tectonic evolution of the Precambrian and Palaeozoic Platforms boundary zone offshore Poland based on the new and vintage multi-channel reflection seismic data

Quang Nguyen[1], Michal Malinowski[1,2], Stanisław Mazur[3], Sergiy Stovba[3,4], Małgorzata Ponikowska[3], and Christian Hübscher[5]

[1] Institute of Geophysics, Polish Academy of Sciences, Warsaw, 01-452, Poland
[2] Geological Survey of Finland, Espoo, FI-02151, Finland
[3] Institute of Geological Sciences, Polish Academy of Sciences, Kraków, 31-002, Poland
[4] S.I. Subbotin Institute of Geophysics, National Academy of Sciences of Ukraine, Kyiv, 02000, Ukraine
[5] Institute of Geophysics, University of Hamburg, 20146, Hamburg

*Correspondence to*: Quang Nguyen (qnguyen@igf.edu.pl)

**Abstract.** The structure of the post-Caledonian sedimentary cover in the transition from the Precambrian to the Palaeozoic Platforms in the Polish sector of the Baltic Sea is a matter of ongoing debate, due to the sparsity of quality seismic data and insufficient well data. The new high-resolution BalTec seismic data acquired in 2016 contributed greatly to deciphering the regional geology of the area. Here we develop an optimal seismic data processing workflow for the selected BalTec seismic profiles offshore Poland. Due to the acquisition in a shallow water environment, the processing strategy focused on suppressing multiple reflections and guided waves, through a cascaded application of SRME, τ-p deconvolution, water bottom F-K filtering, and parabolic Radon multiple elimination. We also perform reprocessing of the legacy PGI97 regional seismic data, achieving significant improvement in overall data quality compared to the original processing. We showcase the potential of the new and reprocessed data focusing seismic interpretation on the area of the Koszalin Fault. In the light of the data available, the Koszalin Fault was the main structure controlling Mesozoic subsidence and Late Cretaceous-Paleocene inversion of the eastern portion of the Mid-Polish Trough offshore Poland. The inversion changed its character from thin- to thick-skinned towards the north, away from the Polish coast. The Koszalin Fault reactivated older structural grain inherited from the time of Devonian continental rifting at the margin of Laurussia. The fault runs obliquely to the CDF, the feature that remained inactive since its formation at the Silurian-Devonian transition.

## 1 Introduction

The transition from the East European Platform to the Palaeozoic Platform of Western Europe has been for a long time a matter of lively discussion (e.g., Berthelsen, 1998; Pharaoh, 1999; Thybo, 2001; Bayer et al., 2002; Mazur et al., 2016; Janik et al., 2022; Ponikowska et al., 2023 and references therein). Besides the Teisseyre-Tornquist Zone (TTZ) which is considered a boundary between the two platforms, there are additional structural elements that contribute to the complexity of the transition





zone such as the Caledonian Deformation Front (CDF; Berthelsen, 1998; Katzung et al., 1993; Lassen et al., 2001; Krawczyk et al., 2002) or late Palaeozoic tectonic grabens (Krzywiec et al., 2022a). These features are particularly apparent in the southern Baltic Sea where the CDF diverges away from the TTZ (Fig. 1A) and a mosaic of various geological blocks, separated by several fault zones, was formed throughout the late Palaeozoic and Mesozoic (e.g., Erlström et al., 1997; Seidel et al., 2018;

Ponikowska et al., 2023). These structures were produced by consecutive tectonic events including, among others, the emplacement of the Caledonian fold-and-thrust belt (e.g., Mazur et al., 2016); late Palaeozoic extension west of the TTZ (Krzywiec et al., 2022a), Mesozoic extension (e.g., Krzywiec et al., 2003; Mazur et al., 2005), and finally Late Cretaceous-Paleocene basin inversion (e.g., Krzywiec, 2002; Krzywiec et al., 2003, 2022b; Al Hseinat and Hübscher, 2017; Pan et al., 2022).

Seismic imaging remains the most effective tool for resolving the complexity of superimposed structural elements in the transition zone from the Precambrian to Palaeozoic Platforms in the southern Baltic Sea. However, sedimentary succession in the Polish sector of the southern Baltic Sea has been so far poorly explored by low-quality industry single-channel and multi-channel seismic (MCS) reflection profiles and a few boreholes (Pokorski, 2010; Jaworowski et al., 2010). The best regional coverage (both in Polish, but also German and Danish sectors) is offered by a high-resolution dataset acquired in 1996-97 in

the southern Baltic Sea (PGI97, Kramarska et al., 1999). These data mostly document the shallow Cenozoic and Mesozoic sedimentary strata (down to 0.8 s TWT) and only in a few places reach the base of the Mesozoic (Krzywiec et al., 2003; 2002). Since the initial interpretation by Krzywiec et al. (2003), no attempts have been made to re-process and re-interpret these data. In March 2016, 3500 km of new MCS profiles were acquired onboard R/V Maria S. Merian (see Hübscher et al., 2017) throughout a large area of the Baltic Sea from the Bay of Kiel to the northeast of Bornholm (Hübscher et al., 2017; Hübscher,

2018) (cruise MSM52, also known as BalTec). Thanks to the unique acquisition parameters, the BalTec MCS data image the tectonic elements from the seafloor down to the Palaeozoic basement. With its large geographic coverage and resolution, these seismic data provide better capabilities for visualizing geological features and mapping structural formations than previous datasets (see examples of new interpretation, e.g. in Ahlrichs et al., 2020; Ahlrichs et al., 2022; Janik et al., 2022; Krzywiec et al., 2022b; Nguyen et al., 2023).

The aim of this contribution is twofold. First, we showcase the development of the optimal seismic data processing workflow for the selected BalTec profiles offshore Poland. A lot of attention was paid to multiple reflection removal strategies for this kind of shallow water data. Furthermore, we investigate if modern processing tools can improve the quality of the legacy PGI97 data and to what extent these data can supplement the interpretations derived from the BalTec profiles. Finally, with the newly processed and reprocessed data, we perform seismic interpretation to put some constraints on the structural pattern

of sedimentary successions in the transition zone from the Precambrian to Palaeozoic Platforms. The central part of the interpretation is the tectonic evolution of the Koszalin Fault and its relation to the CDF offshore Poland.





**Figure 1:** Location of the studied seismic profiles overlaid on main structural elements of the southern Baltic Sea. (A) The bathymetry of the southern Baltic Sea and the location of seismic lines (red lines) were processed and shown in the study. Additionally, the location of the other deep seismic profiles from previous investigations: DEKORP-PQ (PQ2-2 to PQ2-6) in the offshore Bornholm (DEKORP-BASINResearch Group, 1999) and PolandSPAN™ PL1-1100 and PL1-5600 in onshore Poland (Mazur et al., 2015; 2016) are shown in green. (B) Geological map of the southern Baltic Sea without post-Paleocene sediments after Kramarska et al. (1999), Schlüter et al. (1998), Sopher et al. (2016) and Pre-Quaternary map of Bornholm (Hansen and Poulsen, 1977). The position of main faults and tectonic blocks as well as the Teisseyre-Tornquist and Sorgenfrei-



Tornquist Zones are adapted from Seidel et al. (2018). Abbreviations: CDF – Caledonian Deformation Front; EEC – East
European Craton; WEP – Western European Platform; KA – Kamień Anticline; KOA – Kołobrzeg Anticline; MPT – Mid-
Polish Trough; STZ – Sorgenfrei-Tornquist Zone; TTZ – Teisseyre-Tornquist Zone.

## 2 Geological background

The southern Baltic Sea area is located in the transition zone between the Fennoscandia Shield as part of the East European
Craton (EEC) and the West European Platform (WEP). This area is characterized by a mosaic of various geological blocks
separated by several faults and fault zones formed throughout the Phanerozoic (Liboriussen et al., 1987; Vejbaek et al., 1994;
Berthelsen, 1998; Pharaoh, 1999; Thybo, 1999; van Wees et al., 2000). The most prominent tectonic features are the NW-SE
trending Sorgenfrei-Tornquist and Teisseyre-Tornquist Zones (STZ and TTZ), crossing the southern Baltic Sea to the north
and south of Bornholm, respectively (Fig. 1A; Berthelsen, 1998; Pharaoh, 1999). These zones are characterized by major,
often deeply rooted faults that governed subsidence and uplift of major crustal blocks during several tectonic phases in the
Palaeozoic, Mesozoic, and locally, Cenozoic.

At the southwestern margin of the EEC, Proterozoic granites and gneisses are overlain by Cambro-Ordovician sediments of
the Baltica passive margin as penetrated by offshore wells (e.g., Franke et al., 1994; Beier and Katzung, 1999). The closure of
the Tornquist Ocean and the subsequent collision of Baltica with Avalonia during the Late Ordovician led to the formation of
the Caledonian fold and thrust belt (Katzung et al., 1993; Dallmeyer et al., 1999; Katzung, 2001; Torsvik and Rehnström,
2003). In this context, the Cambro-Ordovician successions of the Baltica margin were overthrust by deformed marine
Ordovician sediments (Katzung et al., 1993). The formation of an accretionary wedge is indicated by seismic surveys offshore
Rügen Island (Babel Working Group, 1993; Thomas et al., 1993; Piske et al., 1994; Schlüter et al., 1997; DEKORP-
BASINResearch Group, 1999). In addition, evidence of deformed Ordovician sediments of marine origin was found in several
deep wells on Rügen Island (e.g., Dallmeyer et al., 1999; Katzung, 2001). The northernmost extension of the Caledonian fold
and thrust belt is marked by the CDF, which can be traced from Lolland and Møn (Denmark) to the north of Rügen Island and
farther towards north-western Poland (Fig. 1A; Liboriussen et al., 1987; Erlström et al., 1997). The CDF onshore NW Poland
is marked out by a thin-skinned fold-and-thrust belt involving Ordovician and Silurian mudrock succession of the Pomeranian
Caledonides (Mazur et al., 2016). The oldest sediments there are late Llanvirn and Caradoc in age and they are accompanied
by fragments of the entire Silurian profile, up to the Pridoli (Podhalańska and Modliński, 2006). This is in contrast to the Rügen
area, where only thick (~1500-2000 m) Ordovician sediments have been drilled (e.g., Katzung et al., 1993) including
Tremadoc–Llanvirn sandstones, shales and greywackes that experienced anchizonal metamorphic conditions (Dallmeyer et
al., 1999). Therefore, the Pomeranian Caledonides with their potential offshore continuation are considered part of the
Caledonian foreland basin that was deformed and included in the orogenic wedge during the late stages of Caledonian
shortening (Mazur et al., 2016). Consequently, the CDF separates the undeformed succession of lower Palaeozoic sediments
in the northeast, referred to as the Baltic Basin, from the Caledonian fold and thrust belt in the southwest (Fig. 1A). The latter



forms the basement of the WEP that is unconformably covered by Devonian-Carboniferous and Permian-Mesozoic successions.

The NW-SE trending Koszalin Fault is regarded as the offshore prolongation of the eastern boundary of the Koszalin-Chojnice
Structural Zone (KCSZ). Geological interpretation of the LT-7 deep refraction profile (Guterch et al., 1994) shows the Koszalin Fault as the distal antithetic break-off in the hanging wall (Berthelsen, 1998); other antithetic faults rotate the downfaulted basement and pre-Permian cover southwest of the Koszalin Fault (Antonowicz et al., 1994). Offshore Poland, the Koszalin Fault delimits the boundary between the Darłowo and Kolobrzeg tectonic blocks and it is associated with deformations in the sub-Permian and Mesozoic strata. The strongly localised subsidence occurred mostly in the Triassic and Jurassic periods,
presumably as a result of local strike-slip displacements, transtension, and the creation of small pull-apart basins (Pietsch and Krzywiec, 1996; Krzywiec, 2002). Later, due to NE-SW compression, this entire tectonic system was inverted during the Late Cretaceous/Paleocene time (Antonowicz et al., 1994). The extensional fault system reactivated reversely in the pre-Zechstein basement is accompanied by asymmetric fault-propagation folds generated within the Mesozoic infill (Schlüter et al., 1997; Dadlez et al., 1995; Kramarska et al., 1999; Krzywiec, 2002).

**3 Data and methods**

**3.1 Dataset used in the study**

The BalTec 2D MCS data were acquired in March 2016 onboard R/V Maria S. Merian by the Federal Institute for Geosciences and Natural Resources Germany (BGR) (for details see Hübscher et al., 2017). The main acquisition parameters and an example of a raw shot gather are presented in Table 1 and Figure 2 respectively. Around 850 km of the BalTec data were
recorded in the Polish sector. The location of the selected BalTec profiles is shown in Figure 1B.

The PGI97 2D MCS data were acquired in 1996-1997 as a collaboration of the Polish Geological Survey (PGI) and the Netherlands Institute of Applied Geophysics (TNO) (Kramarska et al., 1999; Krzywiec et al., 2003). This project aimed to image the geology of the upper 400 meters beneath the sea bottom to prepare a new geological map of the Baltic Sea (Kramarska et al., 1999). The dataset consists of several profiles extending to German, Danish and Swedish sectors, with a
total length of 4000 km. The main acquisition parameters and an example of a raw shot gather are shown in Table 1 and Figure 2 respectively. Two profiles were selected to complement the BalTec data for interpretation purposes in this study. Available pre-stack data (raw shot gathers) transcribed from original tapes by PGI make the reprocessing possible.

BalTec and PGI97 data were complemented by the DBE (Danish Bornholm Enclave) dataset, acquired offshore Bornholm in 1982 by Western Geophysical. Geological Survey of Denmark and Greenland (GEUS) provided us with the selected profiles.
Only post-stack time migrated versions were available.



In this study, in order to image the Koszalin Fault, CDF and the sedimentary cover overlying TTZ, 5 profiles were selected from 3 different data vintages (see Fig. 1 for locations): 2 profiles of the BalTec data (BGR16-258, BGR16-212), 2 profiles of the PGI97 data (PGI97-13, PGI97-202) and 1 profile of the DBE data (DBE-6A). The selected sections run from the vicinity of the Polish coast toward the Amager tectonic block in the Danish waters. Most of the seismic sections are oriented from the

northeast to the southwest, and perpendicular to the Koszalin Fault. Unfortunately, well data information is very limited for this study. Only the SW end of profile BGR16-258 is calibrated by deep well L2-1/87 (Fig. 1). The borehole information is limited offshore Pomerania because this area is considered less favourable for hydrocarbon exploration (Karnkowski et al., 2010). However, our seismic interpretation relies on publicly available data from all offshore wells shown in Figure 1 (G14-1/86, Pernille-1, Stina-1, K9-1/86, K1-1/86, L2-1/87, A8-1/83).

| Survey name | | BalTec | PGI97 |
|---|---|---|---|
| General | Recorded by | The Federal Institute for Geosciences and Natural Resources (BGR) | The Netherlands Institute of Applied Geophysics (TNO) |
| | Party/Vessel | RV Maria S. Merian | Dr. Lubecki |
| | Positioning system | DigiCOURSE System 3 | DGPS System |
| | Date | March, 2016 | September, 1996 |
| Seismic source | Type | Airgun | Airgun |
| | No. guns | 8 | 1 |
| | Capacity | 250 cu. in. | 10 cu. in. |
| | Shot interval | 25 m | 12.5 m |
| | Source tow depth | 3/6 m | 1.5 m |
| Receivers | Number of channels | 216 | 12 |
| | Channels interval | 12.5 m | 12.5 m |
| | Cable tow depth | 4 m | 1.5 m |
| | Nearest offset | 32.8 m | 25 m |
| | Furthest offset | 2724.2 m | 162.5 m |
| | Record length | 8500 ms | 800 ms |
| Recording system | Sample rate | 1 ms | 0.5 ms |

**Table 1:** Acquisition parameters for the two seismic surveys: BalTec and PGI97.





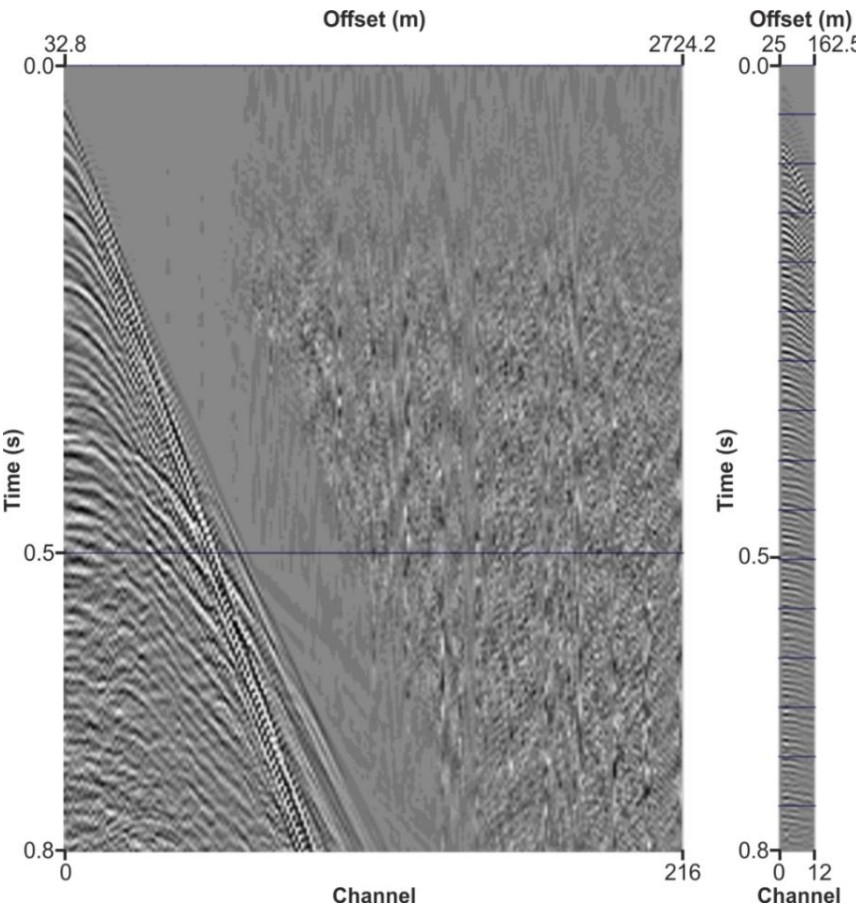

**Figure 2:** Raw shot gathers from the two different surveys (BalTec and PGI97). Notice the large difference in terms of the number of channels between the two datasets.

## 3.2 Multiple reflection elimination strategies

Multiples reflection (multiples) are a constant problem for marine seismic data and an even more challenging issue in shallow waters. The water depth of the southern Baltic Sea is around 10 to 50 m. Therefore, the seismic records in this area will be always contaminated with multiple reflections, especially at near seafloor intervals. There are several ways to classify multiples (Fig. 3). Depending on where the ray path's downward reflection occurs, multiples can be categorized into surface-related multiples and internal multiples. Internal multiples have one downward bounce at the first reflector and no downward bounce

at the surface. Surface-related multiples have two downward bounces at different reflectors: one at the surface and one at the first reflector (Verschuur, 2013). Multiples can be also classified as long-path and short-path based on their time delay from the primary events (Sheriff and Geldart, 1995). Long-path multiples can be separated from the primary events as their travel



path is much longer than the primaries, therefore, they appear at different arrival times in the seismic data. In contrast, short-path multiples are those that cannot be observed as separate events from the primaries that generate them (Verschuur, 2013).

The application of multiple reflection removal (demultiple) methods is dependent on the properties of the multiple presented in the seismic data (Verschuur, 2013). There are two main categories: (1) methods based on a difference in spatial behaviour of primaries and multiples; and (2) methods based on periodicity and predictability. The former is typically based on filtering methods such as τ-p deconvolution, Radon demultiple or stacking, which exploits the differences in velocities and/or reflecting structures between the primaries and multiples. The latter relies on the prediction from either modelling or inversion of the

recorded seismic wavefield (Weglein, 1999). Surface-related multiple elimination (SRME) or wavefield extrapolation methods are among this category (Verschuur, 2013). In our processing workflow, we included both categories of multiple elimination methods in order to handle different multiples existing in the 2D shallow marine seismic data.

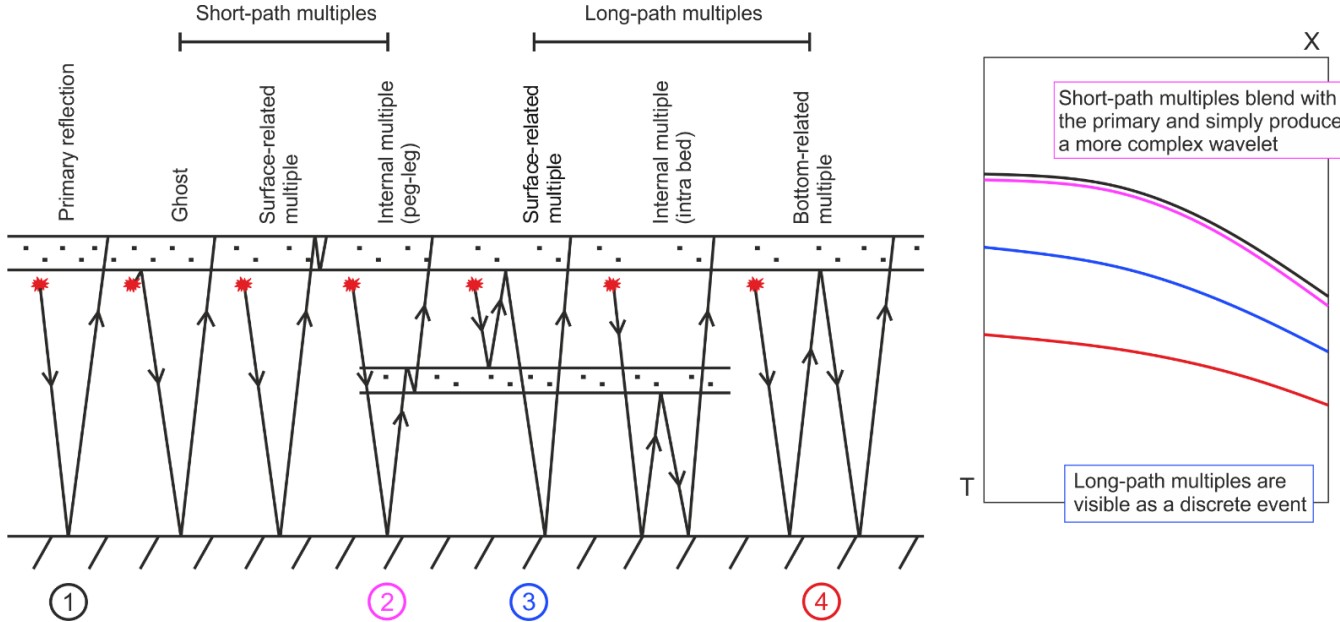

**Figure 3:** Basic types of multiple reflections (modified from Sheriff and Geldart, 1995).

**4 Processing of seismic data**

| BalTec – workflow | |
|---|---|
| 1 | Read SEG-D data and assign geometry |
| 2 | Streamer and shot depth statics |
| 3 | Spherical Divergence |
| 4 | Swell noise attenuation |



| | | |
|---|---|---|
| 5 | | SRME |
| 6 | | Predictive deconvolution in the τ-p domain and τ-p mute |
| 7 | | Data regularisation |
| 8 | | Water bottom F-K filtering |
| 9 | | Velocity analysis |
| 10 | | High-resolution parabolic Radon demultiple |
| 11 | | F-X deconvolution |
| 12 | | Bandpass filter |
| 13 | | Phase conversion |
| 14 | | Pre-stack time migration |
| 15 | | Front mute |
| 16 | | AGC |
| 17 | | Stacking |
| 19 | | Post-stack processing (Trace mixing) |

*PGI97 – original workflow*

| | | |
|---|---|---|
| 1 | | Read SEG-Y data and assign geometry |
| 2 | | Static shift (shot delay) |
| 3 | | Amplitude recovery |
| 4 | | Velocity analysis |
| 5 | | NMO correction |
| 6 | | Stacking |
| 7 | | Post-stack predictive deconvolution |
| 8 | | Bandpass filter |

*PGI97 – new workflow*

| | | |
|---|---|---|
| 1 | | Read SEG-Y data and assign geometry |
| 2 | | Static shift (shot delay) |
| 3 | | Resample |
| 4 | | Source signature deconvolution |
| 5 | | Bandpass filter |
| 6 | | Spherical Divergence |
| 7 | | Pre-stack predictive deconvolution |
| 8 | | Water bottom F-K filtering |
| 9 | | Velocity analysis |





| 10 | Bandpass filter |
| 11 | Front mute |
| 12 | AGC |
| 13 | Stacking |
| 14 | Post-stack time migration |
| 15 | Post-stack predictive deconvolution |
| 16 | Trace mixing |

**Table 2:** Processing workflow applied to BalTec and PGI97 data and the original processing workflow of PGI97 data.

## 4.1 Processing of the new BalTec data

Multiple reflections are the major problem faced when processing seismic data acquired in the Baltic Sea due to the relatively shallow water depth (average water depth around 50 m), and the new BalTec survey is not an exception. Therefore, removing multiples was central throughout the entire processing flow for these data. Based on analysis of the properties of multiples present in seismic data (both in the shot domain and the CDP domain), the demultiple workflow consisted of SRME, water bottom F-K filtering (periodicity and predictably-based methods), τ-p deconvolution, and high-resolution parabolic Radon demultiple (difference in spatial behaviour-based methods). The order of demultiple methods in the processing workflow is shown in Table 2. Seismic gathers in the shot and CDP domain after each demultiple method are compared in Figure 4. A series of stack sections after each demultiple approach are shown in Figure 5.

SRME is a method using a data-driven algorithm for removing multiples (Verschuur et al., 1992), it is often applied earlier than other demultiple approaches in the processing flow. However, SRME's efficiency is largely affected by the signal-to-noise ratio of the input data. Therefore, gun and cable static correction, spherical divergence, and other incoherent noise attenuation approaches should be applied to the SRME input while maintaining the amplitudes and phase of recorded primaries and multiples. The shot point interval is 25 m while the channel interval is 12.5 m. Therefore, to construct the multiple estimates, the shot points were interpolated to 12.5 m, and the record data is extrapolated to zero offsets. A mute is applied to the input shot records prior to removing direct arrival energy and the first seafloor multiple. After surface-related multiple is modelled by a series of convolutions and summation, the output model shot point is renumbered back to the same order as the original shot gathers for the subtracting process. For this data, a three-passes approach for SRME adaptive subtraction including two subtraction processors based on two publications by Wang (2003) and Monk (1993) is applied.

The τ-p predictive deconvolution method is applied following the SRME method. The main aim of τ-p deconvolution for this dataset is to remove short-path multiples. At the same time, muting performed in the τ-p domain attenuates linear noise and seismic interference. The input data is transformed to the τ-p domain using linear transform (Stoffa et al., 1981; Zhou and Greenhalgh, 1994), Wiener deconvolution (design window of 240 ms of operator length and 48 ms of gap length) is applied

 

to the whole traces. After τ-p deconvolution, in order to improve signal-to-noise ratio and enhance deeper imaging data were

regularized in 2 steps. Firstly, the shot records are interpolated to 12.5 m of shot point interval from the nominal 25 m recorded,

which doubled the CDP fold. Secondly, decimating the fold of the shot gathers from 216 to 113 to get to a 25 m receiver

interval and 12.5 m CDP interval so the spatial sampling at 12.5 m is more beneficial to image the deeper part of the sections.

**Figure 4:** Example of shot gathers and CDP gathers after the main processing steps of the BalTec data. A) Raw shot gather;
B) Shot gather after SRME; C) Shot gather after τ-p deconvolution; D) CDP gather after water bottom F-K filtering (input for
Radon demultiple); E) CDP gather after Radon demultiple. The main types of noises and multiples are highlighted by arrows.
Notice how different events are attenuated after each demultiple approach.



Because of the very shallow water bottom, SRME does not successfully eliminate strong water bottom multiples. Therefore, to remove them, we used a simple technique in which the data are flattened according to the travel time of the water-bottom multiple followed by a F-K reject mute to remove flattened multiple energy (see Nguyen, 2020). This procedure (called the water bottom F-K filtering approach) can be repeated for the N-th water bottom multiple. Here we applied up to three passes.

Parabolic Radon demultiple (PRT demultiple) is applied to target the long-path multiples as SRME and τ-p predictive deconvolution methods are inefficient for these types of noises. The normal PRT demultiple has some limitations, so a Harlan signal extraction filter (based on Harlan, 1995) is used to overcome the limitations (this approach was called high-resolution PRT demultiple). This filter provides a high-resolution Radon transform by taking the PRT of the input data, and the muted PRT of the input data with trace polarities randomly reversed and put these into Harlan's signal extraction algorithm to focus on the signal in the PRT domain. The high-resolution PRT demultiple is applied with a start time of 500 ms as artefacts are introduced in the near-surface, where move-out values are large and reflections only exist at near offsets. Optionally, this approach also is reapplied after each velocity analysis pass.

In order to enhance the signal, F-X deconvolution is applied to remove random noise in CDP gathers. F-X deconvolution is a process designed to effectively attenuate random noise by prediction of the non-random signal content in a seismic trace. Each input trace is transformed into the frequency domain. Groups of traces are used to design filters to predict the Fourier components of adjacent traces. The filtered section is finally transformed back into T-X space, and the noise component is removed. NMO correction is applied before and removed after applying F-X deconvolution.

The velocity models were determined using the interactive velocity analysis program (Global Claritas CVA). Generally, velocity analysis was carried out in 3 passes: the first pass velocity is picked before the high-resolution Radon demultiple, the second pass is picked and updated before pre-stack time migration, and the third pass is checked and updated the second pass velocity after migration. Velocity was picked at every 500 CDP point for the first pass, then 250 CDPs (around 3.0 km intervals) for the next two passes. A straight-ray 2D Kirchhoff pre-stack time migration (PSTM) is performed using first-pass velocities to enhance data imaging. The main input of the migration process is pre-stack seismic data with geometry applied and RMS velocity field. The pre-stack migration process is applied again when each pass of the velocity field is updated. Before stacking, a post-NMO outer trace mute is applied to remove any coherent noise on the outer traces and to reduce the effect of NMO stretching on the far offsets. To balance seismic amplitude across the section, a normal 500 ms window AGC scaling process is applied before and after stacking. The traces within each CDP gather were summed. The maximum number of traces in migrated CDP gathers is 108.

More details of the BalTec data processing flow can be found in the BalTec seismic data processing report (see Nguyen, 2020).









**Figure 5:** Part of stack sections of line BGR16-202. A) Raw stack; B) Stack after SRME; C) Stack after SRME and τ-p
deconvolution; D) Stack after SRME, τ-p deconvolution, and Water bottom F-K filtering. Black arrows highlight attenuated
multiples after each demultiple approach. Oval areas show examples of improved reflections.

### 4.2 Reprocessing of the vintage PGI97 dataset

The PGI 97 data were initially processed at TNO, the processing flow can be found in Table 2. The original workflow includes
a limited number of processing steps. Apparently, the original flow is limited to demultiple approaches, especially before
stacking due to constraints on demultiple techniques at that time and the limitations of the data (short offsets).

An example of stacked sections created using the original workflow is shown in Figure 7A. Reverberations from the primary
events are dominant across the seismic section. Because of data acquired in a very shallow water environment, the first and
second order of water bottom multiples are clear and conflict with the primary reflection on the shallow part of the section.
Additionally, the existence of high-frequency noise can be easily seen in the section, which can lead to the misinterpretation
of the main geological features. Secondly, the current sections were stacked using a poor velocity model, which was
represented by the poor image of the water bottom and shallow reflections, also primary events were not continuously imaged.
Lastly, there was no migration process in the original flow, the dipping events are mis-positioned laterally and appear less
steep than they really are, and diffractions events would interfere with primaries. Re-processing addressed all the above issues.

Re-processing workflow for the PGI97 data can be found in Table 2. Since the PGI97 data has a very short offset (only 12
channels) (Fig. 2), the move-out-based or model-based demultiple approaches were not applied to this data. Only the predictive
deconvolution approach was applied before and after stacking. It efficiently attenuates severe reverberations and water bottom
multiples (Fig. 6). Apart from predictive deconvolution, the water bottom F-K filtering technique was also applied to this data
to remove more multiples from the seafloor.

The velocity analysis from the original processing flow did not give the accurate interval velocities due to the length of the
streamer (150 m total active length); therefore, re-picking of velocities was also carried out. Although there were still
uncertainties on velocities due to the limit of traces in the CDPs gathers, the new velocity models were considered relatively
good because they were picked on cleaner gathers and following geology in the study area. The data were then migrated by
post-stack finite different time migration routine (Claerbout and Doherty, 1972) using the new velocity model (Figure 7).





**Figure 6:** Example of shot gathers after main processing steps of the PGI 97 data and their autocorrelation (AC); (A) Raw shot gathers, (B) Shot gathers after FK mute and spherical divergence, (C) Shot gathers after predictive deconvolution, (D) Shot gathers after water bottom F-K filtering. Notice the visible attenuation of reverberation in the AC after the predictive deconvolution.





**Figure 7:** Comparison of part of final stack sections of vintage PGI97-202 profile: unmigrated original (A), unmigrated reprocessed (B), and migrated reprocessed (C) with new BGR16-258 profile at nearby location (see Fig. 1). Black arrows show the noticeable enhancement between the original processing flow with reprocessing flow.

## 5 Seismic interpretation

Seismic interpretation is demonstrated for the selected 5 profiles (BGR16-212, BGR16-258, DBE-6A, PGI97-202, and PGI97-13; Figs. 8-11). As mentioned before, there is very sparse well control in the study area. For this reason, the interpretation of



these five profiles was made together with the profiles shown in Figure 1B, which in turn, were tied to all boreholes in the study area. The horizon interpretation was then extended to other profiles shown in Figures. 8-11 through cross-sections. Southwest of the CDF, horizon identification mainly follows well L2/1-87 stratigraphy markers intersected by line BGR16-258 (Fig. 1B). Previous studies in the south-west of the Baltic Sea close to the Koszalin Fault were also taken into account 270 (e.g., Vejbaek et al., 1994; Krzywiec et al., 2003; Graversen, 2004; Pokorski, 2010; Jaworowski et al., 2010). In the seismic sections, the horizons are interpreted as the base of each formation.

*Profile BGR16-212*

The section of the BGR16-212 profile used in this study crosses the CDF and the Koszalin Fault (Fig 8). The latter appears as a major inversion feature uplifting Jurassic strata to the seafloor surface SW of the fault. The uplifted Jurassic forms a core of 275 the major inversion-related fold, the Kołobrzeg Anticline (Fig. 1B). The anticline represents the offshore continuation of the Mid-Polish Anticlinorium that splits into the Kołobrzeg and Kamień Anticlines NW of the Polish coast (Fig. 1B). The lack of Cretaceous strata suggests deep syn-inversion erosion. The amount of uplift cannot be estimated since the original thickness of the Jurassic and Cretaceous remains unknown. However, the minimum amount of uplift is constrained to 0.7 s TWT (~1 km) by the top of the pre-Carboniferous and top of the Jurassic. The presence of extensive Triassic and Jurassic strata as well 280 as thin Zechstein and Carboniferous layers on the hanging wall of the reversed Koszalin Fault indicates that the latter was originally formed as a normal dip-slip fault (Fig. 8). The minimum downthrown of the hanging wall can be estimated to 1.3 s TWT (~1.7 km), but this amount is only a minimum measure as missing part of Jurassic and entire Cretaceous is not included. Furthermore, the amount of pre-Zechstein erosion remains unconstrained.

The foot wall of the Koszalin Fault is onlapped by a Late Cretaceous syn-inversion marginal trough (Fig. 8). The trough fringes 285 the uplifted core of the Kołobrzeg Anticline, attaining a thickness of 0.7 s TWT i.e., ~1 km. The rim of the Late Cretaceous trough onlaps the Koszalin Fault and the margin of the Kołobrzeg Anticline (Fig. 8). Cretaceous sediments in the footwall are deposited directly on the Silurian substratum. This implies that the footwall was uplifted and eroded throughout the late Palaeozoic and Mesozoic until the late Cretaceous. The exact timing of uplift and erosion cannot be determined from the profile and this issue is addressed in the discussion.

The CDF coincides with the overthrust of the thrust-folded lower Palaeozoic on the mostly undeformed lower Palaeozoic strata of the Baltic Basin. The thrust has not been reactivated during the Late Cretaceous inversion. The amplitude of the overthrust as well as the timing of deformation cannot be inferred from seismic data. Nevertheless, the stratigraphic record (Podhalańska and Modlinski, 2010) points to the Silurian-Devonian transition as the time of folding and thrusting. The Caledonian thrust is of a thin-skinned character (Mazur et al., 2016), a characteristic that is not fully clear from profile BGR16-212. In the profile, 295 the CDF coincides with the basement slope forming a frontal ramp of the thrust system (Fig. 8). SW of the CDF, the top of the basement is located below the extent of the seismic data.





**Figure 8:** Uninterpreted and interpreted part of BalTec line BGR16-212 (see Fig. 1 for location).

*Profile BGR16-258*

There are a number of similarities between profiles BGR16-258 and BGR16-212 (Figs. 8, 9). The Koszalin Fault again appears as a major inversion feature uplifting Jurassic strata to the seafloor surface SW of the fault. At the extreme SW of the profile, the Triassic is exposed at the sea bottom (Fig. 9). The uplifted Triassic and Jurassic form a core of the major inversion-related



Kołobrzeg Anticline (Fig. 1B). The lack of Cretaceous and, in the far SW, also Jurassic strata suggests deep syn-inversion erosion. Only a minimum amount of uplift can be estimated since the original thickness of the Jurassic and Cretaceous remains

unknown. Minimum uplift is constrained to 1 s TWT (~1.3 km) by the top of pre-Carboniferous and top of Jurassic. The presence of extensive Triassic and Jurassic strata as well as Zechstein, Carboniferous and Devonian layers on the hanging wall indicates that the Koszalin Fault was originally formed as a normal dip-slip fault (Fig. 9). The minimum downthrown of the hanging wall can be estimated to 1.3 s TWT (~1.7 km), but this is a minimum amount as missing Jurassic and Cretaceous is not considered. Moreover, two more faults, similar to the Koszalin Fault, occur farther SW. They cause another downthrow of

the top of Silurian by 0.5-1 s TWT. The more southwestern of these two faults is a growth fault indicating Devonian syn-sedimentary extensional tectonics.

A Late Cretaceous syn-inversion marginal trough occupies the footwall of the Koszalin Fault (Fig. 9). The trough developed next to the uplifted Kołobrzeg Anticline, attaining a maximum thickness of 0.8 s TWT i.e., ~1.1 km. The margin of the Late Cretaceous trough onlaps the Koszalin Fault and the adjacent part of the Kołobrzeg Anticline (Fig. 9), suggesting a continuation

of erosion and subsidence sometime after the cessation of inversion. Cretaceous sediments cover a regional top Silurian unconformity proving deep pre-inversion erosion of the present-day foot wall of the Koszalin Fault.

The CDF coincides with the overthrust of the deformed lower Palaeozoic of the Pomeranian Caledonides on the tectonically undisturbed lower Palaeozoic of the Baltic Basin. As in line BGR16-212, there are no indications of the inversion-related reactivation of the Caledonian thrust. The latter is imaged as a relatively sharp boundary separating the orogenic wedge from

the undeformed foreland. A nearby analogue of such a structure is the Carpathian Thrust Front in SE Poland, where a relatively sharp frontal thrust separates deformed sediments of the Carpathian foredeep from their undeformed equivalents (Krzywiec, 2001; Gągała et al., 2012; Krzywiec et al., 2014). The Caledonian thrust branches off from the top of the basement (Fig. 9) the slope of which forms a frontal ramp for the Caledonian fold-and-thrust belt (Mazur et al., 2016). The top of the basement slopes SW-ward below the extent of the seismic data.






**Figure 9:** Uninterpreted and interpreted BalTec line BGR16-258 (see Fig. 1 for location).





*Profile DBE-6A*

This is the NW-most profile out of the five presented in this paper (Fig. 1B). In the DBE-6A line, the CDF is not imaged since
it is located farther west beyond the termination of the seismic profile. This is the case since the CDF diverges away from the
Koszalin Fault turning toward the WNW (Fig. 1B). The Koszalin Fault is imaged again as an important inversion structure
(Fig. 10). Although in a map-scale picture the fault still represents the NE limit of the Kołobrzeg Anticline the seismic profile
reveals a lower order fault-propagation fold within the hanging wall of the Koszalin Fault (Fig. 10). The top of basement is
also involved in folding that proves a thick-skinned character of deformation. The top of the basement is only slightly displaced
and there is virtually no displacement for the top of Silurian. However, higher up, the Devonian to Jurassic strata are juxtaposed
across the fault with the Upper Cretaceous fill of the marginal trough. It means that the original normal, down-dip displacement
in the order of 0.7 s TWT (~0.9 km) was entirely reversed during the inversion phase. A foot wall of the Koszalin Fault is
onlapped by the Late Cretaceous marginal trough that is c. 0.7 s TWT thick (~0.9 km).

Although profile DBE-6A lacks evidence for the end of Silurian and Caledonian deformation, it still documents the late
Palaeozoic-Mesozoic subsidence in the hanging wall of the Koszalin Fault. This is demonstrated by the presence of the
Carboniferous, Zechstein, Triassic, and Jurassic layers. This structural geometry indicates that the Koszalin Fault was
originally formed in the late Palaeozoic as a normal, dip-slip feature. Furthermore, the seismic profile indicates that the
structural domain characterised by late Palaeozoic or Mesozoic, pre-Late Cretaceous uplift and erosion is limited from the W
by the Koszalin Fault and the CDF did not contribute to the development of the top Silurian unconformity in the area farther
east.

An interesting feature represents another fault imaged near the NE termination of the profile (Fig. 10). The fault cuts through
the Darłowo Block (Fig. 1B) from the top of the basement upward into the Late Cretaceous trough. This is clearly an inversion-
related structure since it disturbs most of the Cretaceous succession (Fig. 10). It also causes localised uplift of the top of the
Cretaceous. The unnamed fault in question is a thick-skinned feature displacing the top of the basement by 0.25 s TWT (0.7
km). Furthermore, the side branches jointly form a positive flower structure indicating a transpressional component during the
Late Cretaceous inversion.

*Profile PGI97-13 and PGI97-202*

Profile PGI97-202 is parallel to the BGR16-258 line, running close to the north of it. In contrast, the PGI97-13 profile adjoins
the BGR16-258 line at a high angle, being also oblique to the Koszalin Fault (Fig. 1B). Both PGI97 lines image the Koszalin
Fault quite schematically (Fig. 11). The fault appears as an inversion structure with a reverse displacement in order of 0.2 s
TWT. Both the profiles also document the previous dip-slip kinematics of the Koszalin Fault revealed by the increased
thickness of the Jurassic strata in the hanging wall (Fig. 11). As in the previously described profiles, the footwall of the Koszalin
Fault is onlapped by the Late Cretaceous syn-inversion marginal trough. The PGI97 profiles image its thickness in the range



of 0.7 s TWT. Similar to the remaining seismic lines presented, the marginal trough onlaps the Koszalin Fault (Fig. 11)
providing evidence for erosion and subsidence, continuing for some time after the termination of inversion. A characteristic
feature, especially of profile PGI97-202, is the roughness of the top of the Cretaceous horizon (Fig. 11), suggesting final post-
inversion uplift and erosion before the commencement of Cenozoic sedimentation.



**Figure 10:** Uninterpreted and interpreted GEUS seismic line DBE-6A (see Fig. 1 for location).





**Figure 11:** Uninterpreted and interpreted PGI97 reprocessed seismic lines PGI97-13 and PGI97-202 (see Fig. 1 for location).

**6 Discussion**

**6.1 Processing of the new BalTec seismic data**

The BalTec processing workflow outlined in section 4.1 focused on removing multiple problems because the data was acquired

in a shallow water environment. Figure 4 highlights the main types of noises and multiples existing in the shot and CDP domain. The first demultiple approach used in the processing flow is SRME to suppress the dominant short-path multiples in the shot domain (red arrows in Fig. 4A). These multiples may be peg legs which bounce once or more between the seafloor and the high amplitude reflecting plane below (potentially the top Silurian) (Levin and Shah, 1977), and reverberations of the seafloor and strong reflection events (McGee, 1991). Demultiple techniques such as predictive deconvolution in X-T or τ-p

domain and SRME are typically used for removing these short-path multiples (Sheriff and Geldart, 1995; Verschuur, 2013). Given that the SRME was more effective in the deeper part of the shot gather, it was chosen over the predictive deconvolution in this case. More importantly, the SRME method can model all kinds of surface-related multiples including water layer-related multiples and peg-legs (Verschuur et al., 1992, Verschuur, 2013). The shot gather after SRME shows short-path multiples and surface-related multiples successfully suppressed (Fig. 4B), especially at near-offset due to the sensitivity of the SRME

algorithm in near-offset multiple reconstructions (Qu et al., 2021).

Following the SRME, τ-p deconvolution is another efficient approach for the multiple elimination from the BalTec data. The τ-p deconvolution when performed together with muting in the τ-p domain addressed different types of noise besides multiples, i.e. direct wave, seismic interference, or guided waves (Fig. 4B). The direct waves travel directly from sources to receivers and appear as linear event in the gather (black arrows in Fig. 4B), they are a typical type of noise in any marine seismic dataset.

Whereas the guided waves are commonly found in seismic data from shallow water environments and are most recognized in far offset in the field record (green arrows in Fig. 4B) (Yilmaz, 2001). The guided waves in the shot gather probably represent multiples of the refracted energy. The example of a shot gather after applying τ-p deconvolution (Fig. 4C) and τ-p mute shows a much cleaner gather without these linear coherent noises, and increases the primary event at far offset. The seismic interference (blue arrows in Fig. 4B) was also suppressed by τ-p deconvolution, this type of noise usually happens during a

multi-vessel acquisition. In this case, the seismic interference and primary events could be separated in the τ-p transform domain because of their contradicting dips (Gulunay et al., 2004; Elboth et al., 2017).

The example stack sections from line BGR16-202 display a great improvement after the demultiple workflow (Fig. 5). There is noticeable suppression of noise and multiples (black arrows in Fig. 5A). Furthermore, primary events are enhanced to avoid misinterpretation (examples in the oval areas in Fig. 5B). However, the first and second order of the water bottom multiples

still exist in the BalTec data stack section (black arrows in Fig. 4C) because it is difficult to predict these events using the SRME algorithm owing to insufficient number of traces in this very shallow water depth area. Therefore, the water bottom F-



K filtering approach was applied in the shot domain to eliminate these shallow water bottom multiples. The stack section after this method (Fig. 5D) shows that the dipping primary reflections are no longer interfered with by multiples associated with the water bottom.

The methods based on velocity discrimination might not work properly for the multiples that appeared in the shallow part of the shot and CDP gathers since the move-out difference is not big enough, and the number of traces or samples which are not muted by NMO stretch mute is too sparse. Therefore, in the CDP domain, the high-resolution parabolic Radon demultiple (high-resolution PRT) approach (Harlan, 1995) in the BalTec processing workflow mostly targets the multiples in the deeper part of the CDP gathers (yellow arrows in Fig. 4D) that have larger move-out compared to the primary due to much longer travel time (Hampson, 1986; Sacchi and Ulrych, 1995). The high-resolution PRT approach provided very promising results (Fig. 4E) in cleaning the CDP gather where the NMO-corrected primary events were no longer distracted by large move-out multiples, making velocity picking on such gathers much easier. However, when the data are stacked, there are not that many differences between the data with/without the Radon approach. Removal of these long-path multiples also helps increase signal-to-noise ratios by later stacking process. The cascaded demultiple approach in the BalTec processing workflow

improved the velocity analysis as well as helped to produce the final PSTM stack section optimal for subsequent interpretation.

### 6.2 Reprocessing of the legacy PGI97 data

In case of the PGI97 data, other than the usual linear events that were handled by the F-K filter, the main types of multiples existing in the shot gathers were short-path reverberations and water bottom multiples (Fig. 6B and C). The limited offset of the PGI97 data (~140 m streamer length and 12 channels) (Fig. 2) constrains the application of modern demultiple approaches.

Therefore, simple predictive deconvolution (24 ms gap and 300 ms operator length) in the X-T domain is applied to suppress the reverberations. The shot gathers after the predictive deconvolution approach shows noticeable multiples removed, represented by much cleaner autocorrelation functions (AC in Fig. 6C). The water bottom F-K filtering method proves the efficiency by eliminating first and second-order multiples generated by the water bottom (Fig. 6D).

The reprocessing workflow of the PGI 97 data shows significant improvement in the signal-to-noise ratio, especially in the

final migrated stacked image. A comparison of the final stack sections is shown in Figure 7, where significant uplifts represented by black arrows can be seen in the migrated reprocessed section. There is less high-frequency random noise in the reprocessed section, and the seafloor and shallow sediments are better represented. Since the overall continuity of reflection events is improved, fault displacements are much easier to interpret. Dipping events are imaged at their true locations after migration, deeper than in the original stack. However, despite a lot of effort in reprocessing workflow to suppress multiples,

the remnants of multiples still exist in the data. The profile BGR16-258 stack section (Fig. 7) illustrates a clearer image of the shallow sediments overlying the Koszalin Fault compared to the parallel profile PGI97-202. This proves the remarkably better quality of the newly acquired long-offset data over the legacy short-offset profiles. Still, because of the regional character of



the PGI97 data, it might be worthwhile to process the whole dataset (only a selection of PGI97 profiles was processed in this study).

**6.2 Interpretation**

Two BGR16 profiles show that the CDF is a sharp thrust contact, juxtaposing the Pomeranian Caledonides and the Caledonian foreland basin (Baltic Basin). This thrust was never reactivated during the subsequent geological history. The profiles do not contradict the interpretation postulating the thin-skinned character of the Caledonian fold-and-thrust belt (Mazur et al., 2016), but they do not provide further evidence to confirm this hypothesis. The Caledonian frontal thrust branches off from the

basement slope that may represent a frontal ramp during the end Silurian shortening. However, this cannot be verified because the top of the Precambrian basement is below the range of seismic imaging (Figs. 8, 9) on the hanging wall of the frontal overthrust.

The Koszalin Fault was originally formed due to Devonian extensional tectonics. The BGR16-258 profile shows that the fault was only the north-easternmost one among a number of extensional features downthrowing the top of Silurian toward the SW

(Fig. 9). However, the Koszalin Fault is a structure that marks out the NE limit of this extensional event. Since the late Palaeozoic tectonic grabens are described along strike onshore Poland and in some other localities of the southern Baltic region (Krzywiec et al., 2022a and references therein), the Koszalin Fault may belong to a family of structures resulting from a wide-range tectonic event. This could have been early Carboniferous rifting affecting the passive margin of Laurussia (Smit et al., 2018), a Middle Devonian continental rifting event (Ponikowska et al., 2023) or a combination of both. Close to the Polish

coast (Figs. 8, 9), and onshore Poland (Mazur et al., 2016), the Koszalin Fault is located SW of the CDF. Therefore, a narrow zone of the Caledonian basement (Chojnice Structural Zone) is accessible by wells, whereas farther SW, the top Silurian is downthrown to depths below the interval penetrated by boreholes (Fig. 12A). However, farther north, the Koszalin Fault crosses the CDF and cuts through the Precambrian Platform (Fig. 10).

A recent apatite and zircon low-temperature thermochronology study postulates an important early Carboniferous uplift in the

area of the Baltic Basin, NE of the CDF and Koszalin Fault (Botor et al., 2021). It remains uncertain from the seismic profiles presented whether this uplift was localised along the pre-existing Devonian faults or had a character of long wavelength doming. The latter possibility seems more likely, observing gradual thinning out of the Carboniferous layer toward the NE (Figs. 8, 9). The domal-style uplift must have persisted until the end of the Permian that is shown by the eastward tapering of the Zechstein layer (Figs. 8, 9). Therefore, the top Silurian unconformity in the area of the Baltic Basin, earlier considered a

Variscan unconformity, may represent an early Carboniferous uplift event. Furthermore, the original extent of Devonian and Carboniferous sediments towards the east could be different than today.

The Koszalin Fault was certainly reactivated as a normal dip-slip feature in the Triassic (Fig. 12B). More than 1 km of Triassic sediments and several hundred meters of Jurassic were deposited during continuing Mesozoic subsidence (Figs. 8, 9). The





thickness of the later eroded Cretaceous sediments remains unconstrained. In the Mesozoic, the Koszalin Fault focused nearly

all subsidence, while other faults were almost inactive (Fig. 8).

During the late Cretaceous-Paleocene basin inversion, the Koszalin Fault was reactivated as a major reverse fault (Fig. 12C). The fault limited the eastern limb of the Kołobrzeg anticline and facilitated its uplift exceeding 1 km (Fig. 1). After the activity ceased, the Koszalin Fault was covered by the marginal part of a syntectonic Cretaceous trough, the subsidence of which lasted longer than the inversion. The onlap of Upper Cretaceous sediments may have been related to the bending of the anticline

crest.



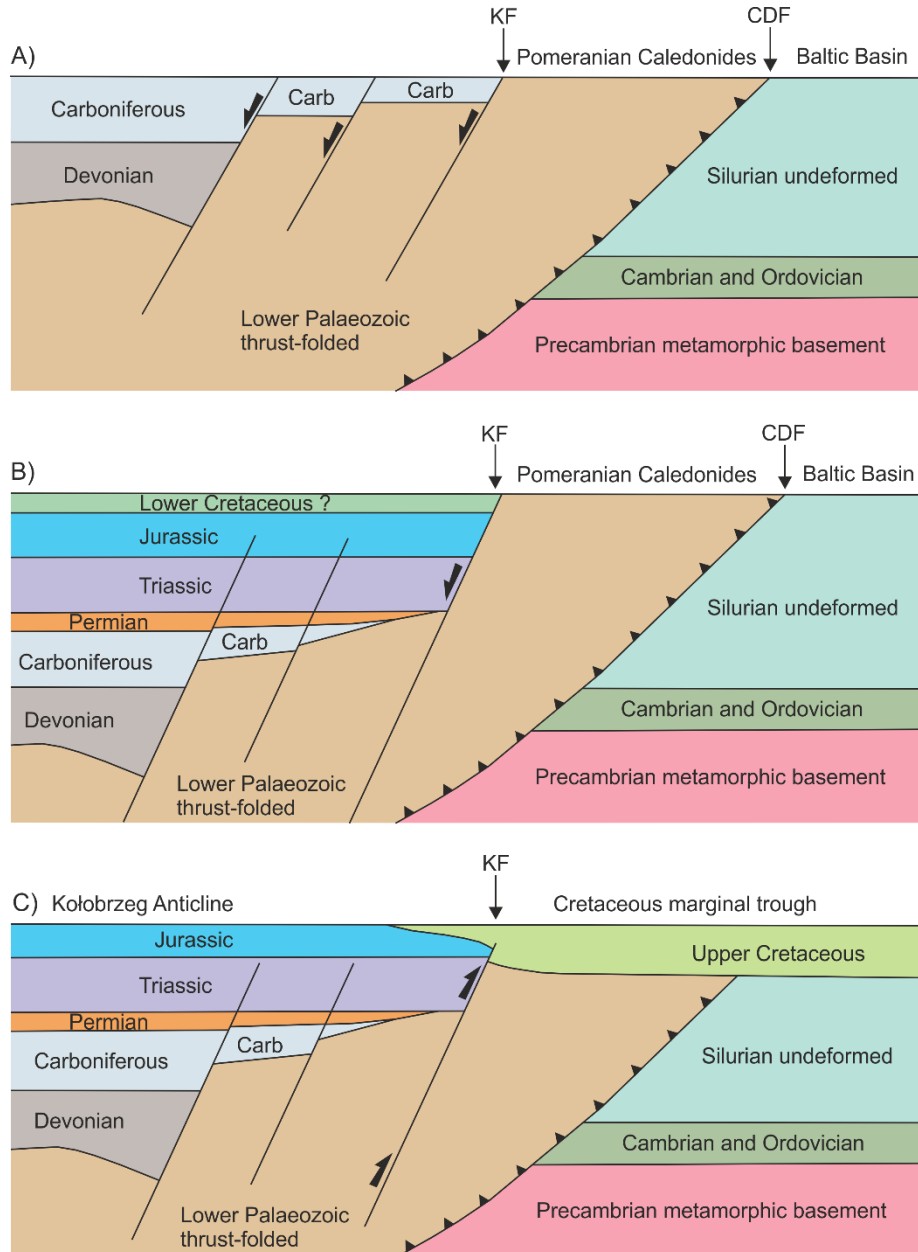

**Figure 12:** Conceptual diagrams showing tectonic evolution of the transition zone between the Precambrian and Palaeozoic Platforms offshore Poland for early Carboniferous (A), middle Cretaceous (B), and post-Paleocene (C). A – Deposition of late Palaeozoic strata west of the Pomeranian Caledonides resulted from normal, dip-slip faulting, initiated in the Middle Devonian and lasting until the early Carboniferous. B – Permian-Mesozoic extension was focused on the Koszalin Fault after a period of late early to late Carboniferous uplift, C – Late Cretaceous-Paleocene inversion formed the Kołobrzeg Anticline and Upper Cretaceous marginal trough.



## 7 Conclusions

In this study, we developed an optimal seismic data processing workflow for the selected BalTec seismic profiles offshore
Poland. Due to the acquisition in a shallow water environment, the processing strategy focused on suppressing multiple
reflections and guided waves, through a cascaded application of SRME, τ-p deconvolution, Water bottom F-K filtering, and
parabolic Radon multiple elimination. The processing workflow presented here could be easily adapted to other multichannel
seismic data acquired in the Baltic Sea (or shallow water in general). We also investigated improving the quality of the legacy
PGI97 seismic data using modern processing techniques. While the reprocessed profiles did not contribute significantly to the
interpretation presented in our study (mostly because of the original acquisition limitations), post-stack migrated results show
significant improvement in overall data quality compared to images from the original processing presented in Krzywiec et al.
(2003). This in turn confirms the value of revisiting legacy data.

With the newly processed and reprocessed data, we attempted seismic interpretation of the sedimentary successions overlying
the crystalline basement in the transition zone from the Precambrian to Palaeozoic Platforms. Even though the interpretation
was hampered by the lack of sufficient well data control, we managed to provide some constraints on the tectonic evolution of
the Koszalin Fault and its relation to the CDF.

The Koszalin Fault was the main structure controlling Mesozoic subsidence and Late Cretaceous-Paleocene inversion of the
eastern portion of the Mid-Polish Trough offshore Poland. The inversion changed its character from thin- to thick-skinned
towards the north, away from the Polish coast. The Koszalin Fault reactivated older structural grain inherited from the time of
Devonian continental rifting at the margin of Laurussia. The fault runs obliquely to the CDF, the feature that remained inactive
since its formation at the Silurian-Devonian transition.

## Acknowledgements

This study was funded by the Polish National Science Centre grant no UMO-2017/27/B/ST10/02316. Cruise MSM52 has been
funded by the German Science Foundation DFG and the Federal Ministry of Education and Research (BMBF). We thank the
Federal Institute for Geosciences and Natural Resources (BGR) for their support during seismic data acquisition and sharing
of the data. DBE data were provided thanks to the agreement with the Geological Survey of Denmark and Greenland (GEUS).

We would like to thank IHS Markit Ltd. and Petrosys Ltd. for the donation of academic licenses of Kingdom Suite and Globe
Claritas software packages, respectively.





**Data availability**

BalTec (MSM52) seismic data processed in this study are available upon request. PGI97 data are owned by the Polish State Treasury and are available via request from the Polish Ministry of Environment. Danish data are available via request from GEUS.

**Author contribution statement**

**Quang Nguyen**: Conceptualization, Methodology, Data Curation, Writing – Original Draft preparation, Software,
Visualization. **Michal Malinowski**: Supervision, Project administration, Writing – Review & Editing. **Stanisław Mazur** and **Sergiy Stovba**: Data Curation, Writing – Original Draft preparation. **Małgorzata Ponikowska**: Visualization, Data Curation. **Christian Hübscher**: Project administration, Data provider.

**Competing interests**

The co-author Michal Malinowski is a member of the editorial board of the Solid Earth journal.

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
