# Peer review of "Post-Caledonian tectonic evolution of the Precambrian and Palaeozoic Platforms boundary zone offshore Poland based on the new and vintage multi-channel reflection seismic data"

_EGUsphere, 2023_

## Referee Comment (RC1)

Dear Editor,

Please, find enclosed my review for the manuscript entitled „Post-Caledonian tectonic evolution of the Precambrian and Palaeozoic Platforms boundary zone offshore Poland based on the new and vintage multi-channel reflection seismic data " by Q. Nguyen et al.

The presented study deals with seismic reflection data from the Baltic Sea in a region, where the Koszalin Fault crosses the Caledonian Deformation Front. The geological framework is quite complicated and the tectonic history still under debate.

To use seismic lines from different acquisition campaigns is a common practise in geophysics as this often enables to profit from a relatively good data coverage. As seismic lines from different acquisition campaigns, which took place up to more than 25 years ago, are used to decipher the tectonic evolution in this part of the Baltic Sea, the goals of this study are not only tectonic ones, but also methodological ones. Whereas the theme of this study (or both themes) fit well in the scope of the journal Solid Earth, the twofold goals are a bit problematic in terms of the manuscript. In its present version, in my opinion, neither goal is lastly achieved, however, the manuscript provides the impression that the focus of the study/manuscript lastly is on the tectonic evolution of the study area in the region of the Koszalin Fault crossing the Caledonian Deformation Front.

Thus, I would strongly recommend to the authors to think about how to clearly express the main aim of the manuscript. One possibility would be to provide details of the processing schemes and multiple removal in supplementary online material, which then would also make space for a more thorough tectonic discussion and e.g. a 4D conceptual tectonic evolution concept. Another way to solve this problem could be to split the manuscript into two manuscripts, part 1 and 2, with the first methodological one focusing on processing (and how to consistently integrate the seismic profiles of various origin) and the second focusing on the tectonic evolution of the study area. One of the reasons for this suggestion is that in my view, in section 5, it is not becoming clear, how the new processing applied to the various seismic lines led to their improvement and added to the presented interpretation.

However, there are statements, which may lead to some confusion: e.g. the last sentence of the „conclusions" sounds as if the fact that the Koszalin Fault runs oblique to the Caledonian Deformation Front is a new finding - however, this is already visible from Figure 1, and thus should be a well known feature. Please clarify and think about the 4D evolution concept, as mentioned in the preceding paragraph. In my view, this is important, as the CDF ist described as an inactive feature, and thus e.g. motions along the Koszalin fault (and other faults) should be relative to the intercepting CDF.

Basically, the manuscript is well organised and figures are both, necessary and helpful. The English clearly would benefit from shaping by a native speaker. Figures should be checked to avoid potential confusion like e.g. in figure 1: annotations of the seismic lines shown in the maps are not consistent, e.g. is the prolongation of DBE-6A shown in red in A and in green in B, also the CDF is not shown in B. Due to the many bright colours, figure 1B is not easy to read.

Drill holes are quite sparse in the study area. However, the authors mention that the interpretation is tight to wells positioned on or close to the seismic lines. Thus, it would be very useful to show such a seismic line together with the stratigraphic record of the drill hole used to tie the interpretation.

In section 4.1, velocity analysis is described. And velocities should be also known from cited wide angle seismic data. Thus, depth migration should be possible, which could be very useful for reconstructing the tectonic evolution of the study area. Would this be an attempt to aid interpretation (and obtain the correct geometry of faults)?

Summarising, in my view, the results of this study could become of interest to be published in Solid Earth, however, only after the mentioned mainly manuscript strategic issues are resolved.

I hope, my comments are of help for your final decision about this manuscript.

---

## Referee Comment (RC2)

[referee-annotated manuscript omitted]

---

## Author Comment (AC1)

**Dear Editor and Referees,**

**Thank you very much for evaluating our manuscript and helpful suggestions. Please find below our responses to all the comments. Our responses are written in bold font. We refer to line numbers in the revised manuscript where changes are highlighted (annotated version of the manuscript). We have also made some corrections to English use and typos throughout the whole text. We think that we have addressed all of your comments, and we hope that the new version might be considered suitable for publication.**

**On behalf of the authors,**

**Quang Nguyen**

*Reviewer #1:*

1. I would strongly recommend to the authors to think about how to clearly express the main aim of the manuscript. One possibility would be to provide details of the processing schemes and multiple removal in supplementary online material, which then would also make space for a more thorough tectonic discussion and e.g. a 4D conceptual tectonic evolution concept. Another way to solve this problem could be to split the manuscript into two manuscripts, part 1 and 2, with the first methodological one focusing on processing (and how to consistently integrate the seismic profiles of various origin) and the second focusing on the tectonic evolution of the study area.

**We decided to move a detailed description of seismic processing to Supplementary Material. The latter was further developed according to recommendations by Reviewer 2 (e.g., by showing velocity models).**

One of the reasons for this suggestion is that in my view, in section 5, it is not becoming clear, how the new processing applied to the various seismic lines led to their improvement and added to the presented interpretation.

**Processing has paramount importance for seismic interpretation and therefore we emphasized this part in our paper. We wanted to demonstrate that our processing of the new BalTec data offshore Poland is the first and the only existing one that produced final PSTM results. For the vintage PGI97 data, the main aim was to check to what extent the reprocessing can improve the quality of the seismic sections. We do not focus though, on how the improved seismic sections can be used in reinterpretation of these data in general. However, we believe that the reprocessing was essential to be able to interpret seismic structures in the vicinity of the Koszalin Fault, which is central to our paper.**

2. However, there are statements, which may lead to some confusion: e.g. the last sentence of the "conclusions" sounds as if the fact that the Koszalin Fault runs oblique to the Caledonian Deformation Front is a new finding - however, this is already visible from Figure 1, and thus should be a well known feature. Please clarify and think about the 4D evolution concept, as mentioned in the preceding paragraph. In my view, this is important, as the CDF is described as an inactive feature, and thus e.g. motions along the Koszalin fault (and other faults) should be relative to the intercepting CDF.

**Thank you for pointing this out. Indeed, the last sentence of 'Conclusions' was unnecessary and has been removed from the revised manuscript. We also appreciate the suggestion to build a 4-D evolution concept, but at this stage, a 4-D model would represent an over-interpretation due to the rather low data density.**

3. Basically, the manuscript is well organised and figures are both, necessary and helpful. The English clearly would benefit from shaping by a native speaker. Figures should be checked to avoid potential confusion like e.g. in figure 1: annotations of the seismic lines shown in the maps are not consistent, e.g. is the prolongation of DBE-6A shown in red in A and in green in B, also the CDF is not shown in B. Due to the many bright colours, figure 1B is not easy to read.

**Thank you for your comment. We fixed the annotations, added the CDF to Figure 1b in the revised manuscript, and modified Figure 1b to make it easier to read.**

4. Drill holes are quite sparse in the study area. However, the authors mention that the interpretation is tight to wells positioned on or close to the seismic lines. Thus, it would be very useful to show such a seismic line together with the stratigraphic record of the drill hole used to tie the interpretation.

**Seismic to well tie is important for the interpretation. Unfortunately, in our study, well data available are limited to just stratigraphy markers and the checkshot survey (time-depth chart). No wells log or stratigraphy records are available. We already mentioned this limitation in section 3. Therefore, the interpretation was based on a few wells (ex. L2-1/87, K9-1/89, or A8-1/83...) and published studies (we mentioned that in section 6).**

**To further mitigate the effect of a limited borehole database, we have added a new figure (Fig. 5) to the revised manuscript to illustrate well ties. All wells contributing to the seismo-stratigraphic interpretation through line intersections are shown in Figure 1b and mentioned in the text.**

5. In section 4.1, velocity analysis is described. And velocities should be also known from cited wide angle seismic data. Thus, depth migration should be possible, which could be very useful for reconstructing the tectonic evolution of the study area. Would this be an attempt to aid interpretation (and obtain the correct geometry of faults)?

**The wide-angle seismic data are too sparse to create a meaningful depth migration model, especially in terms of lateral velocity variations within individual formations. As soon as depth-migrated seismic data are shown, readers assume that, for example, all undulations of layer thicknesses are geologically real. With time sections, readers know that imaging artifacts are possible. We are convinced that inaccurate depth sections are more misleading than time sections. We hence stick to time migrated sections.**

---

## Author Comment (AC2)

**Dear Editor and Referees,**

**Thank you very much for evaluating our manuscript and helpful suggestions. Please find below our responses to all the comments. Our responses are written in bold font. We refer to line numbers in the revised manuscript where changes are highlighted (annotated version of the manuscript). We have also made some corrections to English use and typos throughout the whole text. We think that we have addressed all of your comments, and we hope that the new version might be considered suitable for publication.**

**On behalf of the authors,**

**Quang Nguyen**

*Reviewer #2:*

Main comments

1. The first part of the work describes the application of a specific workflow to the BalTec data acquired in 2016, but it does not provide new information on the processing compared to the report published in 2020 (Nguyen Q, 2020, Seismic Data Processing Report (BALTEC / MSM52). Institute of Geophysics PAS. https://dspace.igf.edu.pl/xmlui/handle/123456789/112). Perhaps the authors could add more details about the processing in this work. For instance, velocity models used for pre-stack migration are not displayed, and it would be helpful to show them alongside the seismic sections. In the description of the re-processing of PG197 profiles, the authors explain the need for a new velocity analysis compared to the original processing and that this new velocity analysis is picked on cleaner CDPs gathers (L251-252); I suggest showing an example of CDPs gather with its corresponding velocity analysis. The authors could also add these figures in a supplementary material.

**We would like to note that our processing work for the new BalTec data is the first and only existing that produced final PSTM results, there was no processing work on this dataset before. The cited processing report is published just internally at IG PAS. We thought some of the developments for processing shallow water seismic data (especially the demultiple strategy) deserve a proper journal publication. Reviewer 1 requested us to remove details on seismic processing, as they were somehow diluting the main focus of the paper (seismic interpretation). We therefore decided to move a detailed description of data processing to Supplementary Material.**

**Following your comments on the velocity models, we included RMS velocity models of two BalTec profiles (lines BGR16-212 and BGR16-258) in Supplementary Figures S4 and S5. Additionally, a figure including velocity analysis and velocity model for one of the PGI97 profiles has been added (Fig. S6).**

2. I would suggest a reorganization of some sections of the paper:

In my opinion, a "results" paragraph is missing, where the outcomes of the processing are illustrated, including a description of seismic sections in terms of reflectivity and characteristics of the reflectors (e.g., amplitudes, frequencies, geometries, continuity, etc.). Additionally, in this paragraph, it would be useful to discuss an estimate of the difference in vertical resolution between the BalTec and PG197 data.

There are several repetitions between paragraphs 4.1 and 6.1; please check.

The "discussion" paragraph could incorporate the geological interpretation of the seismic sections, by merging the content of the current paragraphs 5 and 6.3.

**In response to the above recommendations, we have reorganized the structure of the manuscript by adding a new "Result" section (Section 5 in the revised manuscript). This section comprises information on reflection patterns, characteristics of the reflectors, and resolution as an introduction to the subsequent interpretation section. Moreover, we added a new figure (Figure 3 in the revised manuscript) to illustrate the efficiency of our in-house processing of the BalTec data compared to the onboard processing.**

**Further details requested by the Reviewer, are now included in Supplementary Material. We moved processing details there to avoid repetitions between Sections 4 and 6.**

**We would like to keep the "Discussion" and "Interpretation" sections separate as they serve different functions in the manuscript.**

3. The authors point out that data interpretation poses a challenge due to the limited well control in the area (L265). In their interpretation, they rely on well L2/1-87 situated at the southwest edge of profile BGR16-258. The horizon interpretation is then expanded from other wells using cross-sections, which are not shown in the manuscript. To enhance their approach, the authors might consider utilizing the boreholes intersected by the other BalTec lines to establish a correlation between stratigraphic markers and seismic facies. Subsequently, they could use the seismic facies to extend the interpretation. This "seismic facies table", together with the description of the seismic facies could represent the new paragraph 5 "seismic interpretation".

**We have added a new figure (Figure 5 in the revised manuscript) that shows 3 wells with stratigraphic markers associated with the seismic sections. We also added a description of the reflection patterns (Section 5 of the revised manuscript) for each geologic formation imaged.**

**We believe that the term "seismic facies" is not appropriate for our study because "seismic facies" is often associated with depositional environments in sequence stratigraphy studies. Instead, we use the term "reflection patterns" to describe reflectivity characteristics.**

4. I would like to preface by stating that seismic interpretation is not my primary expertise, and I hope that other reviewers can provide insights on this matter. I found the interpretation of the seismic sections to be insufficiently detailed, often overlooking clear structures and containing some errors. I recommend that the authors revisit the interpretation, aiming to better follow the main reflectors indicated by the seismic imaging and interpreting details that, in my opinion, could support the tectonic conceptual model. Furthermore, I recommend enhancing the line drawing by following the geometries of the reflectors. I have highlighted some examples in the attached PDF.

**Following the Reviewer's recommendations, we have made the seismic interpretation more detailed to some extent, as much as the data allowed. In many cases, however, the change is not significant. We also checked all the elements specifically indicated by the reviewer in the annotated manuscript. Additionally, we have slightly updated the range of seismic profiles presented to avoid unclear**

**locations that are not sufficiently documented in the data (see revised seismic profiles in Figures 6 and 7). We believe that our interpretation effectively substantiates the main findings of our research.**

5. The authors have reprocessed the PG197-13 and PG197-202 profiles, demonstrating the enhancement between the original processing flow and the reprocessing flow in Figure 7. I agree with the improvement in data quality. However, the interpretation of these reprocessed profiles is less detailed compared to the interpretation of the original profiles proposed by Krzywiec et al. 2003. Improving vintage data should aim to achieve a more detailed interpretation of the data. Therefore, I suggest that the authors also reconsider the interpretation of the reprocessed PG197 profiles.

**We would like to note that the interpretation part of the manuscript focuses on the Koszalin Fault zone. The interpretation from Krzywiec et al. (2003) shows a completely different story and is unrelated to the Koszalin Fault. We mentioned this study in the manuscript because this is the only peer-reviewed paper in which PGI97 data were used.**

**We do admit that the PGI97 data do not play an important role in the Koszalin Fault area interpretation due to the associated acquisition limitations. Nevertheless, we would still like to present the improvement in the quality of the data from this survey obtained by reprocessing using modern tools.**

6. Specific comments and minor suggestions are provided in the attached PDF file.

**Thank you for these comments. We took them into account when revising our manuscript. The changes introduced are visible in the tracked version of the manuscript.**